

# Hydrometeorological Data from Baker Creek Research Watershed, Northwest Territories, Canada

Christopher Spence and Newell Hedstrom

Environment and Climate Change Canada, Saskatoon, SK Canada

Correspondence to: Christopher Spence (chris.spence@canada.ca)

It is uncommon to collect long term coordinated hydrometeorological and hydrological data in northern circumpolar regions. However, such datasets can be very valuable for engineering design, improving environmental prediction tools or detecting change. This dataset documents physiographic, hydrometeorological and hydrological conditions in the Baker Creek Research Watershed from 2003 to 2016. Baker Creek drains water from 155 km$^2$ of subarctic Canadian Shield terrain in Canada's Northwest Territories. Seasonal half hourly hydrometeorological data were collected from representative locations, including exposed Precambrian bedrock ridges, peatlands, open black spruce forest and lakes. Hydrometeorological data includes radiation fluxes, rainfall, temperature, humidity, winds, barometric pressure, and turbulent energy fluxes. Terrestrial sites were monitored for ground temperature and soil moisture. Spring maximum snowpack water equivalent, depth and density data are included. Daily streamflow data are available from a series of nested watersheds ranging from in size from 9 to 128 km$^2$. These data are unique in this remote region and provide scientific and engineering communities with an opportunity to advance understanding of geophysical processes and improve infrastructure resiliency. The data described here are available at: https://www.frdr.ca/repo/handle/doi:10.20383/101.026



## 1 Introduction


The subarctic Canadian Shield is typified by a landscape dominated by exposed Precambrian bedrock, numerous lakes, open forest and wetlands. It is a large region, occupying 1.26 million $km^2$, and 13% of Canada. It extends from the Northwest Territories in the west, and east through the Nunavut territory, the provinces of Saskatchewan, Manitoba, Quebec, and Newfoundland and Labrador. By its sheer size, it represents an important region for

Canada. It is rich in mineral deposits, including gold, diamonds, rare earth elements, nickel, and copper. There are several active mines across the region, but also many abandoned mines that represent a significant environmental liability. Hydroelectric power generation is a significant economic activity. The region produces an estimated 10% of the freshwater yield that flows towards the Arctic Ocean (Prowse and Flegg, 2000), so the hydrometeorology of this region and how it may or may not be changing has implications for more southerly latitudes.


The ability to make informed decisions about engineering design and predict short and long term predictions about environmental conditions in this region can be hindered by a lack of data. The large areas and limited transportation infrastructure make intensive data collection expensive. There are only 2 long term climate stations in the region that presently collect precipitation data adequate for long term change detection (Laudon et al., 2017). It is rare to

have multiple gauges on streams that can provide data to glean predominant runoff and streamflow processes. Such information is important also for developing and testing water and energy cycling algorithms used in both weather and climate models.

To help fill this gap, the Baker Creek Research Watershed was instituted in the early 2000's. The

hydrometeorological data produced from activities in this watershed include that describing atmospheric conditions, soil climate and streamflow among a series of nested watersheds. These data are the only integrated hydrological and hydrometeorological small basin dataset in the western subarctic Canadian Shield, and so represent a significant dataset for the region useful for baseline characterization. The data described here represent a valuable resource of hydrometeorological conditions that can be used for engineering design, improving environmental prediction tools

or detecting change. While these data are not representative of the conditions in locations further afield in the subarctic Canadian Shield they can be used to understand hydrological processes, and inform environmental model structure, that can be used across the region. Cautious use of some of the data with longer periods of record could be used in change detection studies.

## 2 Site description

The Baker Creek Research Watershed is located in the subarctic Canadian Shield landscape of Canada's Northwest Territories near the city of Yellowknife. Baker Creek is a stream characterized by lakes connected by short channels that drains water from ~165 $km^2$ into Great Slave Lake (Figure 1). The observation focus is in the upper reaches of the stream, specifically that area of the watershed upstream of the Water Survey of Canada (WSC) hydrometric gauge Baker Creek at the outlet of Lower Martin Lake (07SB013), which has a topographic drainage area of 155



km$^2$. There are 349 perennial lakes in the basin, many small, and the median and mean lake areas are 5,400 m$^2$ and 88,800 m$^2$, respectively.   The basin is in the zone of discontinuous permafrost.  Large changes in topography, vegetation, winter snow accumulation, local hydrology and surficial geology over short distances result in abrupt transitions from permafrost to non-permafrost conditions.  Glaciolacustrine clays, outwash and organic deposits are typically underlain by permafrost, whereas bedrock and well-drained glaciofluvial sands are typically unfrozen

(Morse et al., 2016).  The thickness of organic soils or fine-grained materials derived from glaciolacustrine sediments, glaciofluvial sands or outwash range from less than 1 m to more than 10 m, depending on underlying bedrock topography.  Well and ground thermistor string installations associated with this dataset rarely involve drilling greater than 10 m before encountering bedrock.  An organic soil layer of about 0.25 m is ubiquitous over fine-grained soils.  Any forest canopy that grows in these areas is typically quite open.  Predominant vegetation

includes black spruce (*Picea mariana*), jack pine (*Pinus banksiana*), paper birch (*Betula papyrifera*), Labrador tea (*Ledum groenlandicum*), moss (*Sphagnum* spp.), and lichen (*Cladonia* spp.).

The Meteorological Service of Canada (MSC) climate station Yellowknife A can be used to characterize a regional climate which has short cool summers (July daily average temperature of 17°C) and long cold winters (January daily

average temperature of -27°C).  Annual unadjusted precipitation averages 289 mm, with 40% of that falling as snow.  Annual snow cover begins in October and lasts until the end of April and beginning of May (Spence et al., 2010).  In most years, the largest input of water to the basin is during the spring freshet (Spence et al., 2011) and the hydrological regime of the basin is described best as subarctic nival.  The average annual streamflow at the outlet of Lower Martin Lake is 0.24 m$^3$/s or 47 mm/a, providing an annual runoff ratio of 0.17.  The runoff regime exhibits

remarkable variation for a basin with almost 350 lakes with a standard deviation in annual streamflow of 0.18 m$^3$/s or 37 mm/a and annual runoff ratios that range from 0.005 (2015) to 0.34 (2001).  A maximum daily streamflow of 8.7 m$^3$/s has been observed, but there are common prolonged periods of zero flow below the outlet of Lower Martin Lake.

**3 Physiographic Data**

Physiographic data available for the Baker Creek Research Watershed include a satellite derived land cover classification at 10 m resolution (Figure 1).  Land cover was mapped using a maximum likelihood supervised classification of a composite image of two SPOT5 MS satellite images collected on May 24, 2008 and June 20, 2009.  Four multispectral bands and the normalized difference vegetation index of both images were used as input information.  Following classification, a mode based filter with a 3 cell by 3 cell mask was passed over the image to

filter out errors commonly experienced at edges due to mixed pixel signatures.  The accuracy of the land cover classification was evaluated using a random sample of 314 points recorded during field surveys and marked with a handheld global positioning system (GPS) accurate to within ±8m.  The overall accuracy of the land cover map was 86% and the kappa coefficient was 0.82.  Land cover was classified into six primary types (Table 1), of which exposed bedrock and lakes/ponds comprise 61% (Phillips et al., 2011).  Variation in soil properties among the

different soil covered land cover types is listed in Table 2 (Guan et al., 2010).



Elevation data (Figure 1) were collected during a LiDAR survey on August 21, 2007.  During the survey, one second interval GPS base station data were collected from Natural Resources Canada's Canadian Active Control Site GPS site in nearby Yellowknife.  Laser point positions were computed by differentially correcting all base

station data, integrating these with the airborne GPS and inertial measurement unit (IMU) to differentially correct the airborne trajectory, and calibrating survey data over known targets.  Data post processing included examining for mismatches between flight lines, and isolating high and low laser pulse returns that were well above the canopy or below the ground surface.  Ground returns were classified from the point cloud within Terrascan (www.terrasolid.com) assuming at least one ground point was within a 400 m$^2$ area, the terrain angle was less than

88°, the iteration angle was less than 6°, and the iteration distance was less than 1.4 m to the plane.  Data sets were adjusted to account for local geoidal undulation.  Orthometric heights and geoid separation were determined using the HTv2.0 model (CGG2000 Scientific model + HRG01 Corrector Surface, allowing the direct transformation of NAD83 or ITRF ellipsoidal heights to CGVD28 orthometric heights).  Ground only points were gridded to 1 m node spacing using 2$^{nd}$ power inverse distance to power functions within a 15 m search radius.  This 1 m dataset was

degraded to 10 m to align with the land cover data described above.  Both the elevation and land cover datasets are projected to NAD1983 UTM Zone 11N (Spence et al., 2010).

## 4 Hydrometeorological Data and Conditions

There are two long term primary climate towers operating in the catchment (Table 3).  These occupy locations above an exposed bedrock ridge and a lake (Figure 1).  As noted above, these two cover types represent the majority of the

catchment area (61%).  They are also arguably the most difficult surfaces over which to estimate turbulent energy fluxes, so both stations are equipped with eddy covariance instrumentation.  Meteorological variables measured at each tower include air temperature, relative humidity or vapour pressure; net radiation and its components, wind speed, barometric pressure, and rainfall.  The tower specifications, including variables and units, sensor types and heights, and surface characteristics are summarized in Table 4.  The eddy covariance instruments measure wind

speed and water vapour content at 10 Hz and fluxes calculated over a half hour period.  Corrections to the eddy covariance measurements include coordinate rotation (Kaimal and Finnigan, 1994) the WPL adjustment (Webb et al., 1980), sonic path length, high frequency attenuation and sensor separation (Massman, 2000; Horst, 1997) and oxygen extinction.  All meteorological data are observed every 5 seconds, and half hourly averages (or totals) are logged on Campbell Scientific 23X or CR3000 data loggers.  The data set includes these half hourly values.  The

period of record of the bedrock ridge tower extends from 2005 to 2016.  The lake tower's period of record is from 2008 to 2016.  There are numerous gaps in the data due to sensor malfunction, none of which were filled.  The stations typically operate from early April to late September each year and are not operated through winter because data integrity cannot be guaranteed because frost may form on the towers because of infrequent site visits.  This frost either prevents batteries from being charged, or sound conditions for data collection.




The strong seasonality of radiation and energy exchanges in this landscape are evident from the climate tower on the exposed bedrock outcrop near Vital Lake (Figure 2). Incoming long wave radiation remains relatively consistent month to month compared to incoming short wave radiation, the latter peaking near 250 W/m$^2$ in June, decreasing to <100 W/m$^2$ by September. Net long wave radiation is almost always negative from exposed bedrock surfaces

during the snow-free season. Short wave albedos are typically low (0.12) except when snow is present (~0.7). Net radiation rises from monthly averages of 75 W/m$^2$ in April to 140 W/m$^2$ in June and then steadily decreases to 50 W/m$^2$ by September. Bowen ratios are high in this dry location, almost always exceeding unity, with monthly averages commonly near 3.0. Only during exceptionally wet autumn periods (2008) were monthly Bowen ratios less than one. At the tower located on Landing Lake, the seasonal cycle begins with predominantly negative Bowen

ratios at the end of winter, as sensible heat fluxes are directed to the snow and ice covered surface. Monthly average latent heat fluxes are always higher than sensible heat fluxes, but the latter do rise relative to the former at the very end of the open water season as conditions get cool and damp (Figure 3). Short wave albedos stay high (0.5) until lake ice break-up in early to mid-June after which they average 0.15.

The meteorological data presented in Figures 4 and 5 do not always fully demonstrate the strong annual cycles of temperature that occur in this region, but monthly average air temperatures usually rise above 0°C in May, peak just below 20°C in July, average 8°C in September, and drop below freezing again in October. Mean annual air temperature in the basin from 2005 to 2016 was -3.4°. Relative humidity rises through the warm season averaging 58% and 74% in July and September, respectively. Monthly wind speeds consistently average between 3 and 4

m/s.

The period from 2005 – 2016 represents an almost complete wet-dry cycle. The nearby Environment and Climate Change Canada meteorological station Yellowknife A, with data beginning in the 1940's, provides excellent context for the data collected at Baker Creek, with an average April – September precipitation of 180 mm. The gauges in

Baker Creek averaged a total of 147 mm over these months from 2005 to 2011, but this decreased 27% to 107 mm between 2012 and 2016. This drought was most severe in 2015 with only 70 mm of rainfall. The data from Yellowknife A implies the period 2013-2015 was the driest three year period since records began in the 1940's. This drought was a major contributor to forest fires in 2014 that were the most extensive in two decades (http://www.cbc.ca/news/canada/north/worst-forest-fires-in-30-years-cost-n-w-t-55m-1.2770136).

**5 Spring Snowpacks**

Each year since 2003 (except 2006) a land cover stratified spring maximum snow pack survey has been conducted within the Baker Creek basin at the locations listed in Table 3. These are typically done in one day within the first two weeks of April. Within each of the five land cover types described above – coniferous forest, deciduous forest, lakes, exposed bedrock and wetlands – is at least one designated 25 point snow course (Figure 6). Wetlands and

peatlands are combined into one cover type for the purposes of the spring snow survey. At each course 25 depths are recorded at least 10 m apart using an aluminum rod marked every 1 cm. At each fifth depth measurement a



density measurement is taken using an ESC-30 snow corer. From these measurements, spring maximum snowpack depth (cm), snow water equivalent (SWE) (mm) and snow density (kg/m$^3$) are estimated for each land cover type. A basin-wide average is calculated by pro-rating these estimates by land cover fractions provided in Table 1. The

accuracy of such a snow survey is expected to be within 15% (Pomeroy and Gray, 1995)

Results indicate the basin-wide spring maximum snow pack averaged 80±17 mm SWE from 2003 – 2016 (Figure 7). Snow depths are shallow, reflecting the dry climate, averaging 38±8 cm. The deepest snowpacks are typically in the wetlands, perhaps as the shrubs that proliferate there experience less interception than the forest and better trap

the snow (Sturm et al., 2001), and thinnest and most homogenous over lakes as these tend to be quite open and wind-blown (Rees et al., 2006; Derksen et al., 2006). The exposed bedrock almost always has the most variable snowpack as the snow is transported from open areas into depressions across the uneven topographic surface (Figure 6).

### 6 Streamflow Data

The nested hydrometric network in the Baker Creek basin (Table 3) includes one Water Survey of Canada gauge and five others operated by the Science and Technology Branch of Environment and Climate Change Canada. The Water Survey of Canada gauge, Baker Creek at the outlet of Lower Martin Lake (07SB013) has been operating since 1983. These streamflow data are available from https://wateroffice.ec.gc.ca/search/historical_e.html. The outlet of Lower Martin Lake often remains open, and the stream cross section is bedrock and extremely stable. This

results in a nearly complete (95%) record of high quality streamflow data that is a foundational component of research catchment operations.

The next oldest hydrometric gauge is at the outlet of Landing Lake (Figure 1), which was installed in 2003 for the Mackenzie GEWEX Study (Woo, 2008). This was followed by Baker Creek at Vital Narrows in 2005, Baker Creek

above Vital Lake and Moss Creek at the outlet of Lake 690 in 2008 and Baker Creek at the outlet of Duckfish Lake in 2009. These last three gauges were initially installed to support Canada's contribution to the International Polar Year. Permanent bench marks were installed in 2017 that allow stage at all the gauges (save Baker Creek above Vital) to be referenced to elevations in meters above sea level. These benchmark locations are referenced to NAD83, but unlike the elevation dataset use orthometric heights in CGVD2013. Any future exercise that merges

the water and topographic elevations must take this into account.

A combination of lower flows, beaver activity and icing-susceptible cross sections at these gauge sites higher in basin make it more difficult to obtain continuous accurate stage measurements, stable rating curves and streamflow estimates. Of the non-WSC gauges, the Landing and Vital Narrows stations are meant to be capable of continuous

stage measurements, while the others are seasonal as stage recorders are removed in late fall to avoid freezing of the transducers. This precludes many instances of known zero-flow periods from appearing in the record. This has recently been resolved with upgrades in instrumentation and siting in 2017 at Baker Creek above Vital, at Duckfish



Lake and in Moss Creek at the outlet of Lake 690. These non-continuous records make it difficult to calculate and compare annual or multi-year basin yields, but the data are useful for hydrological model testing and process studies

(Spence et al., 2006; Spence et al., 2010; Phillips et al., 2011; Spence et al., 2014; Spence et al., 2018).

The long Water Survey of Canada dataset from Baker Creek at the outlet of Lower Martin Lake provides context for the shorter records of the upstream gauges. The period beginning in 2003 was characterized by typically average to above average streamflow in the watershed, reflective of precipitation inputs (Figure 4 and Figure 7). The high

rainfall sometimes was heavy enough to generate late warm season runoff events (Spence et al., 2011; 2014) (e.g., 2008, 2011) that exceeded spring freshet events. These late fall and early winter events could not be captured at all gauges in all years because of icing conditions, and are more evident at the Vital Narrows and Landing Lake gauges which stay ice free longer under high water conditions. The meteorological drought noted earlier (Figure 4 and Figure 7) that began in 2013 also manifested into hydrological drought (Figure 8). This effect is most pronounced at

the highest gauge, at Duckfish Lake, which has not experienced outflow since 2014 as water levels remain below the lake outlet elevation. This kind of hydrologic disconnectivity is common across the watershed during dry conditions, and this can reduce streamflow so much that lake evaporation along the watercourse can prevent water from areas actively producing runoff from proceeding downstream (Spence 2006, Spence et al., 2018). This was particularly evident in 2015 and 2016 with the differences in streamflow at the Vital Narrows, Landing Lake and

Lower Martin gauges. During these types of periods, the sub-catchments of the Baker Creek watershed behave as individuals, rather than part of a cohesive watershed.

**7 Ground Temperature and Soil Moisture Data**

Shallow ground temperatures and soil moisture have been measured since 2007 at locations listed in Table 3. Decagon Em50 $ECH_2O$ data loggers with Decagon 5TM capacitance soil moisture and temperature sensors at the

ground surface (i.e., 1 cm depth) and 25 cm depth are deployed at five locations throughout the basin. These sites represent the different typologies and topologies of areas with overburden; peat filled bedrock depressions, forested hillslopes below bedrock ridges, and wetlands between lakes (Table 2). These data have been used to estimate storage changes in these cover types across the basin assuming that variability among such different land cover types is larger than that within (Spence et al., 2010). Half hourly data are typically available from April – December

capturing both spring thaw and fall freeze-back (Figure 9). The sensors have been prone to animal disturbance, so data completeness is uncommon. The sensors do not measure total water content, but total unfrozen water content, so values are unavailable through winter, and the soil moisture time series often captures an increase in unfrozen water during spring thaw, and a decrease during freeze-back. This should not be interpreted as a change in storage, but a change in phase. The active layer depth varies among the sites, often shallowest (0.3 m) in wetlands and

deepest in the peatlands (~1.5 m) (Guan et al., 2010). Soils near the surface often reach saturation during spring snowmelt, but then quickly dry (Figure 10). The dry climate means the elevation of the water table tends to follow the descent of the frost table in the drier hillslopes, with commensurate decreases in unsaturated soil moisture. Soil moisture at depth in wetlands between lakes is much more stable than the hillslopes, likely because of moisture



distributed from runoff travelling through the wetland. Moisture distribution in these wetlands is known to be
uneven (Spence et al., 2011b), and edges of these wetlands may be actually better represented by measurements
taken in forested hillslopes. The water table in peatlands is often near the surface, and the soil column is often near
saturation through the entire warm season.

To calculate ground heat fluxes within the footprint of the bedrock ridge tower, a shallow 0.5 m borehole was drilled
in 2007 and instrumented with HOBO pendant thermistors initially at the surface, 20 cm, 30 cm, and 46 cm, and
only from April – September each year. In 2010, the thermistors were changed to HOBO Pro v2 soil/water probes
and one at 10 cm was added. These sensors provide half hourly data to an external logger that permits year round
measurements. The high thermal conductivity of the granite in this location results in much warmer ground
temperatures than in areas of overburden. The temperature of the shallow bedrock often exceeds 25°C in July, while
unfrozen soil columns remain closer to 12°C (Figure 11). Conversely, in the middle of winter, these data imply
shallow depths in bedrock outcrops can be as cold as -17°C, while soil columns do not get colder than -10°C.

## 8  Data availability

In partnership with Portage, Compute Canada, and the Canadian Association of Research Libraries, the University
of Saskatchewan launched the Federated Research Data Repository (FRDR), which is a single online location from
which research data can be shared, preserved, discovered, curated, and cited. The data described here are available
at: https://www.frdr.ca/repo/handle/doi:10.20383/101.026.

## 9 Summary

These data represent an effort to measure all components of the energy and water cycle in a catchment in the
subarctic Canadian Shield landscape. The period of record for some components extends from 2003. The goal of
making the data available to the research and applied hydrology communities is twofold. First, it is meant to support
and inform water resource management decision making. The record is an important source of baseline data that
can be used to assess the effect of disturbance, such as fire or resource development. Second, these data are
provided to allow others to also investigate hydrological processes and medium-term patterns and potential changes
in this extensive landscape. Measurements have employed consistent methods to ensure comparability within the
research catchment and perhaps eventually permit time series analysis. The data has proven fit for the purpose of
supporting hydrological and hydrometeorological process research and this can ensure proper interpretation of
spatial and temporal patterns and causal attribution.

## Declaration of competing interests

The authors declare that they have no conflict of interest.



**Acknowledgements**

There have been several people over the years that have contributed to data collection, and deserve recognition. These include Shawne and Steve Kokelj, Meg McCluskie, and Stefan Goodman, Scott Dowler and Ryan Gregory from the Government of the Northwest Territories; May Guan, Ross Phillips, and Kirby Ebel from the University of Saskatchewan; and Cuyler Onclin, Mark Russell and Kerry Pippy from Environment and Climate Change Canada

Water Science and Technology Directorate.  The Yellowknife office of the Water Survey of Canada has provided outstanding logistical support for the research catchment; thank you to Jason Friesen, Bob Reid, Derek Forsbloom, Rick Klakowich and Dale Ross.

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



**TABLES**


Table 1: Land cover percentages from SPOT-5 classification exercise.

| Cover | km$^2$ | % |
|---|---|---|
| Coniferous forest hillslopes | 31.88 | 20.78 |
| Deciduous forest hillslopes | 1.15 | 0.75 |
| Exposed bedrock | 61.24 | 39.92 |
| Peatland | 15.45 | 10.07 |
| Water | 34.72 | 22.63 |
| Wetland | 8.98 | 5.85 |

Table 2: Soil characteristics in each typical soil-covered landscape type present in the Baker Creek Research Watershed. K denotes hydraulic conductivity.

| Trait | Peatland | Forested Hillslope | Wetland |
|---|---|---|---|
| Porosity | 0.85 | 0.83 | 0.8 |
| Bulk density (kg/m$^3$) | 78 | 113 | 104 |
| Particle density (kg/m$^3$) | 574 | 644 | 567 |
| Specific yield | 0.15 | 0.19 | 0.25 |
| K (0-0.5 m) (m/s) | $10^{-6}$ | $10^{-5}$ to $10^{-7}$ | $10^{-6}$ to $10^{-7}$ |
| K (> 0.5 m) (m/s) | $10^{-7}$ to $10^{-8}$ | $10^{-8}$ to $10^{-9}$ | $10^{-6}$ to $10^{-9}$ |




Table 3: List of monitoring locations in the Baker Creek Research Watershed

| Site | Land cover | Latitude | Longitude |
|---|---|---|---|
| **Snow courses** | | | |
| Vital Lake | Water | 62.6087° | -114.4489° |
| Ryan Lake | Water | 62.5868° | -114.3709° |
| North wetland | Wetland | 62.6529° | -114.4800° |
| Landing portage | Coniferous forest hillslope | 62.5483° | -114.4051° |
| Jack pine above Vital | Coniferous forest hillslope | 62.6242° | -114.4534° |
| Outcrop above camp | Exposed bedrock | 62.5937° | -114.4387° |
| Outcrop near Ryan Lake road | Exposed bedrock | 62.5689° | -114.3639° |
| Landing Lake east shore | Deciduous forest hillslope | 62.5631° | -114.4001° |
| **Climate towers** | | | |
| Vital upland | Exposed bedrock | 62.6042° | -114.4475° |
| Landing Lake | Water | 62.5593° | -114.4117° |
| **Hydrometric gauges** | | | |
| Baker Creek at outlet of Landing Lake | | 62.5499° | -114.4005° |
| Baker Creek at Vital Narrows | | 62.5792° | -114.4159° |
| Outlet of Lake 690 | | 62.5946° | -114.4436° |
| Baker Creek above Vital Lake | | 62.6190° | -114.4545° |
| Baker Creek at outlet of Duckfish Lake | | 62.6476° | -114.4477° |
| **Ground temperature and moisture nests** | | | |
| Landing portage | Coniferous forest hillslope | 62.5483° | -114.4051° |
| Wetboot peatland | Peatland | 62.5728° | -114.4071° |
| Camp valley | Coniferous forest hillslope | 62.5938° | -114.4375° |
| Tower Peatland | Peatland | 62.5998° | -114.4425° |
| Wetland above Vital | Wetland | 62.6190 | -114.4545 |
| Vital upland | Exposed bedrock | 62.6042 | -114.4476 |




Table 4: Sensor suites at the primary climate stations at the bedrock ridge and Landing Lake.

| Variable | Bedrock ridge | | Landing Lake | |
| | Sensor | Height (m) | Sensor | Height (m) |
| --- | --- | --- | --- | --- |
| Air temperature / Relative humidity | Vaisala HMP45C | 2.8 and 4.4 | Vaisala HMP45C | 1.4 |
| Surface temperature | n/a | | Apogee infra-red thermometer | 4.0 |
| Turbulent fluxes | Campbell Scientific CSAT-3 and LiCor LI-7500 | 4.7 | Campbell Scientific CSAT-3 and KH20 | 3.2 |
| Incoming short wave radiation | Kipp and Zonen CNR1 | 4.2 | Li-Cor pyranometer | 3.5 |
| Incoming long wave radiation | Kipp and Zonen CNR1 | 4.2 | n/a | |
| Outgoing short wave radiation | Kipp and Zonen CNR1 | 4.2 | Li-Cor pyranometer | 0.95 |
| Outgoing long wave radiation | Kipp and Zonen CNR1 | 4.2 | n/a | |
| Net radiation | | | Kipp and Zonen NRLite | 1.65 |
| Wind speed | Met One 013A | 4.4 | Met One 013A | 1.1 |
| Barometric pressure | Li-Cor Li-7500 | 4.7 | RM Young 612 | 1.5 |
| Rainfall | Texas Instruments TE525M | 5.3 | n/a | |
| Wind Direction | n/a | | NRG systems | 4.3 |
| Vegetation height | | 3.9 | | n/a |
| Displacement height | | 2.6 | | n/a |
| Roughness height | | 0.5 | | n/a |



**FIGURE CAPTIONS**

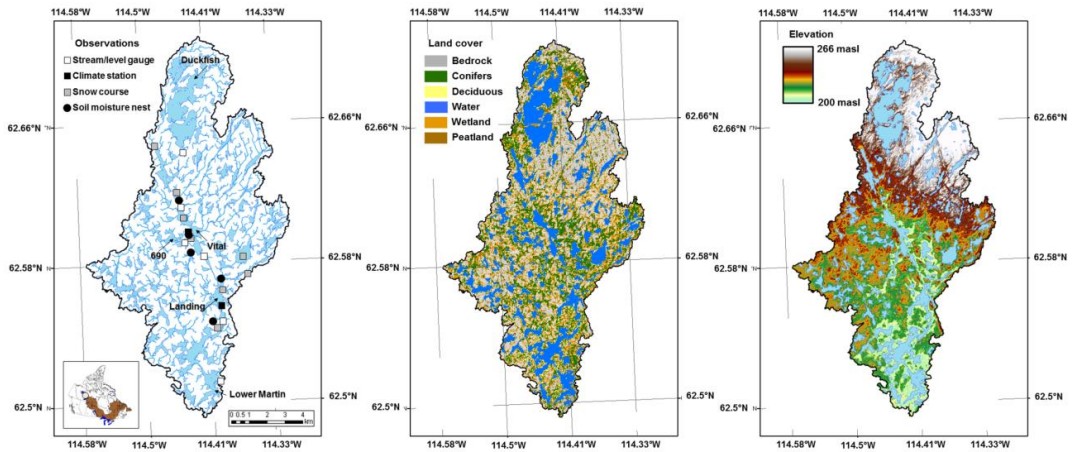

Figure 1:  Three maps of the Baker Creek watershed illustrating left to right; the drainage network (derived from
1:50,000 scale National Topographic System maps) and observation sites; land cover; and elevation.  Lakes that are
monitored are named in the left panel.  The outlet of the watershed defined by the maps is at Lower Martin Lake.
The inset map of the Northwest Territories shows the Canadian Shield ecozones, and Baker Creek's location by the
grey circle.





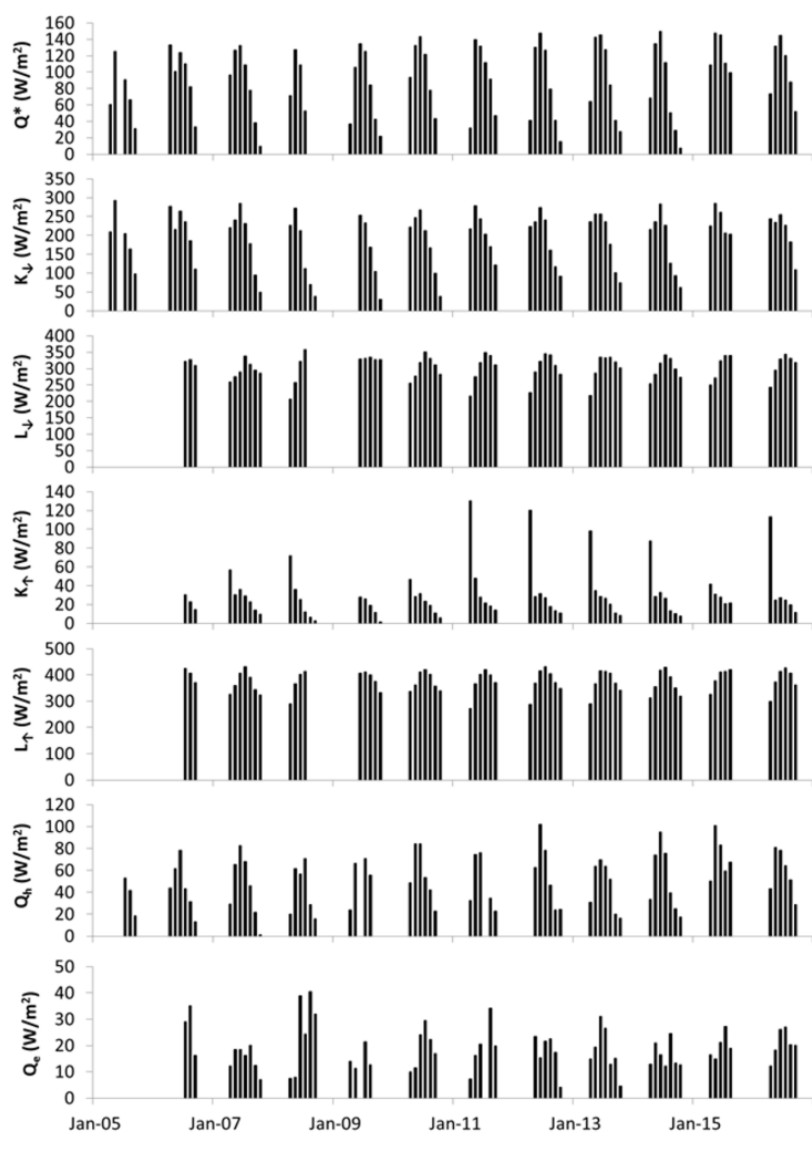


Figure 2: Monthly average radiation and energy fluxes at the bedrock ridge climate tower from 2005 - 2016. All terms in are in W/m². The radiation terms, Q*, K↓, K↑, L↓ and L↑ are net radiation, downwelling short wave radiation, upwelling short wave radiation, downwelling long wave radiation, upwelling long wave radiation, respectively. The turbulent energy fluxes are sensible heat, $Q_h$, and latent heat, $Q_e$.





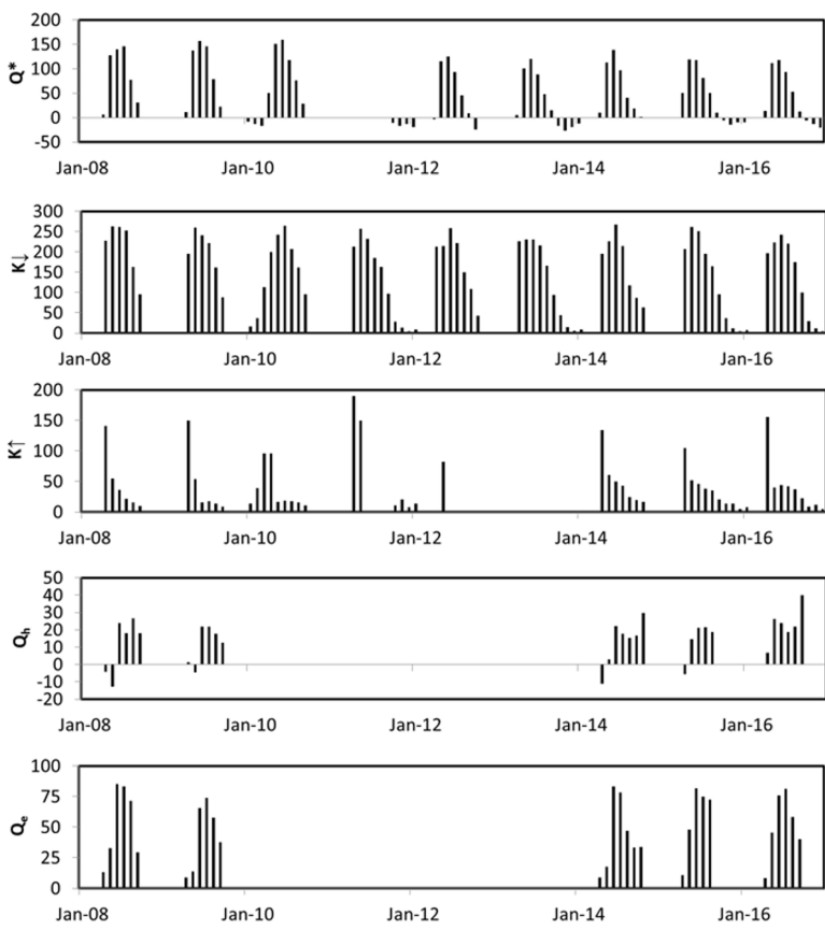


Figure 3: Monthly average radiation and energy fluxes at the Landing Lake climate tower from 2008 – 2016. All terms are in W/m$^2$. Nomenclature is identical to that in Figure 2.




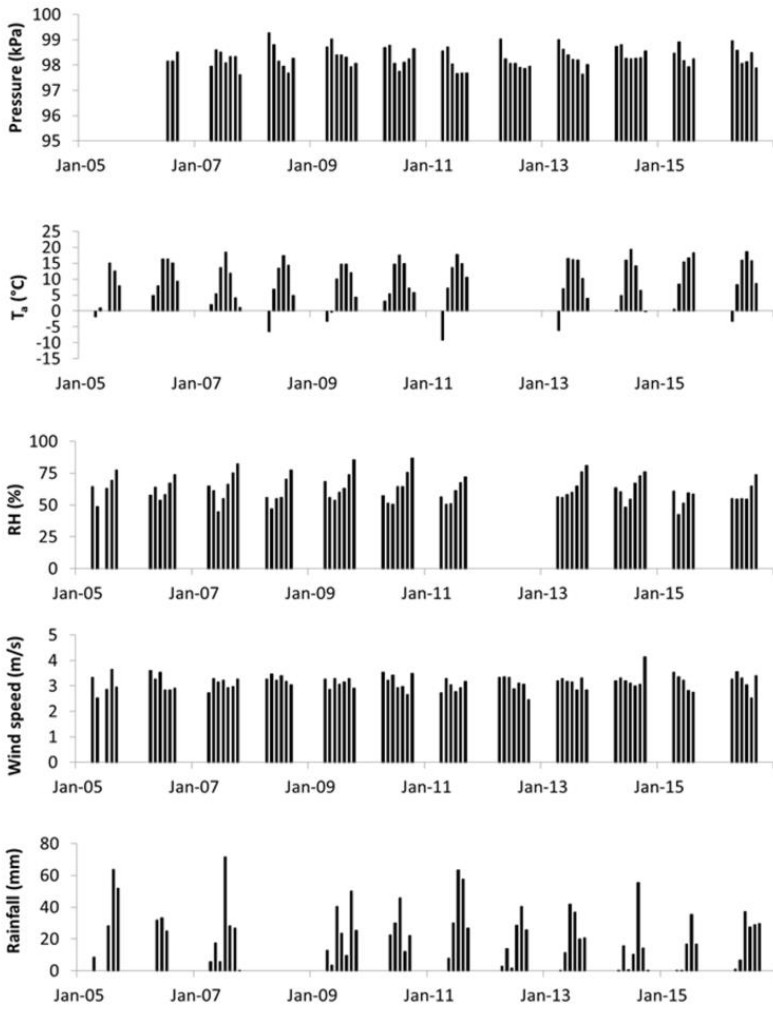

Figure 4:  Monthly average meteorological conditions at the bedrock ridge climate tower from 2005 – 2016.  $T_a$
refers to air temperature, RH denotes relative humidity, and pressure refers to barometric pressure.

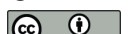



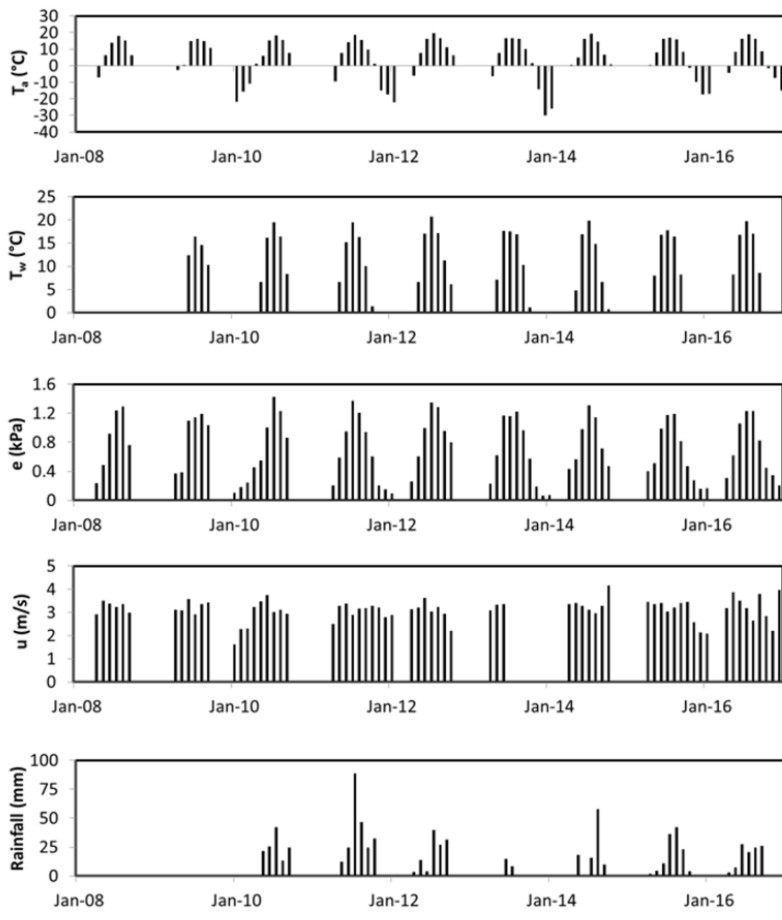

Figure 5: Monthly average meteorological conditions over Landing Lake from 2008 – 2016. $T_a$ and $T_w$ denote air and surface water temperature, respectively. Vapour pressure is e, and u is wind speed.




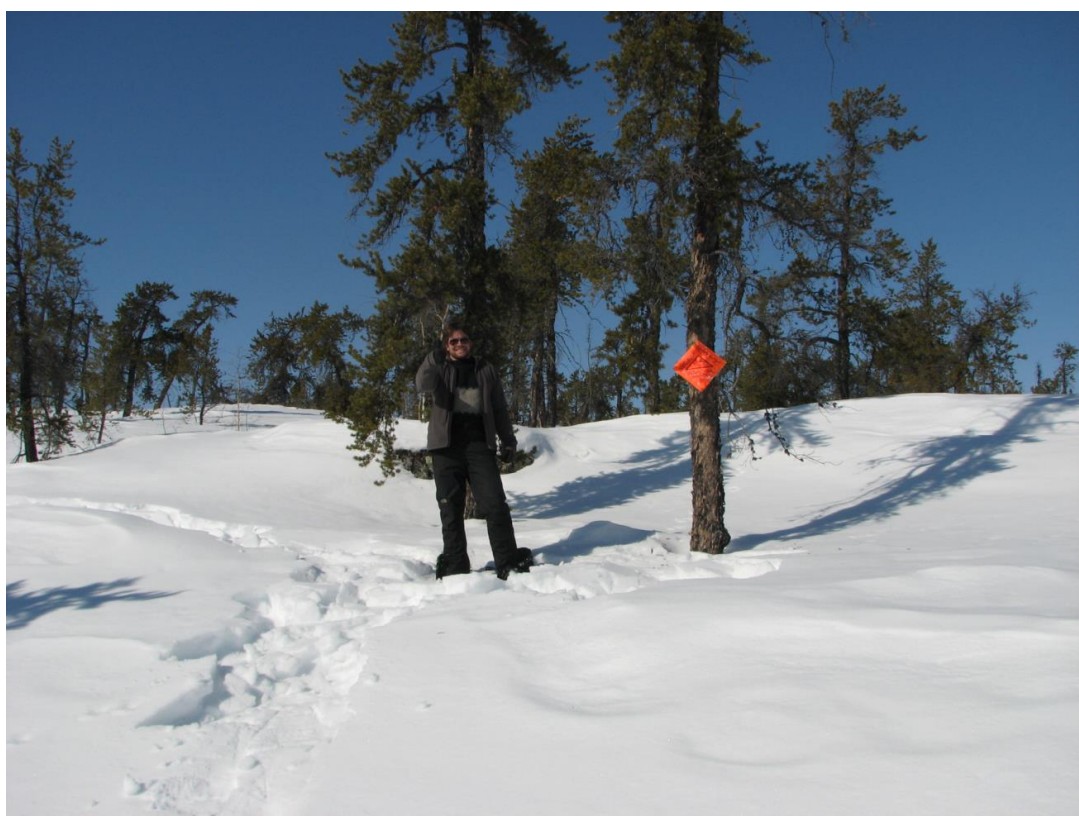

Figure 6: The beginning of one of the exposed bedrock snow courses. Note the sparse jack pine individuals and exposed rock behind Ross Phillips.





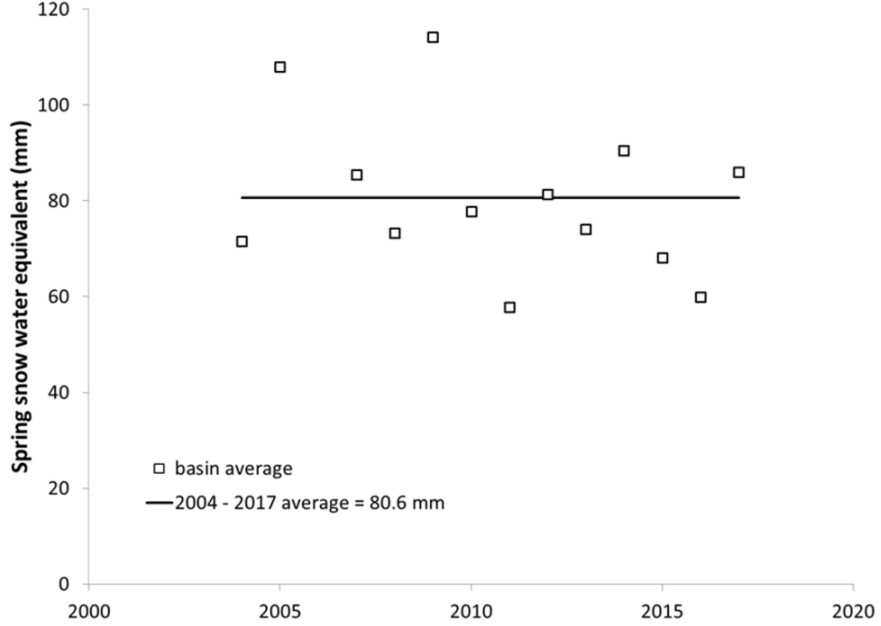


Figure 7: Basin average spring maximum snow water equivalent from 2004 – 2017.  The period of record average is included as the solid horizontal line, which highlights the trend of declining snowpacks during the current precipitation cycle.



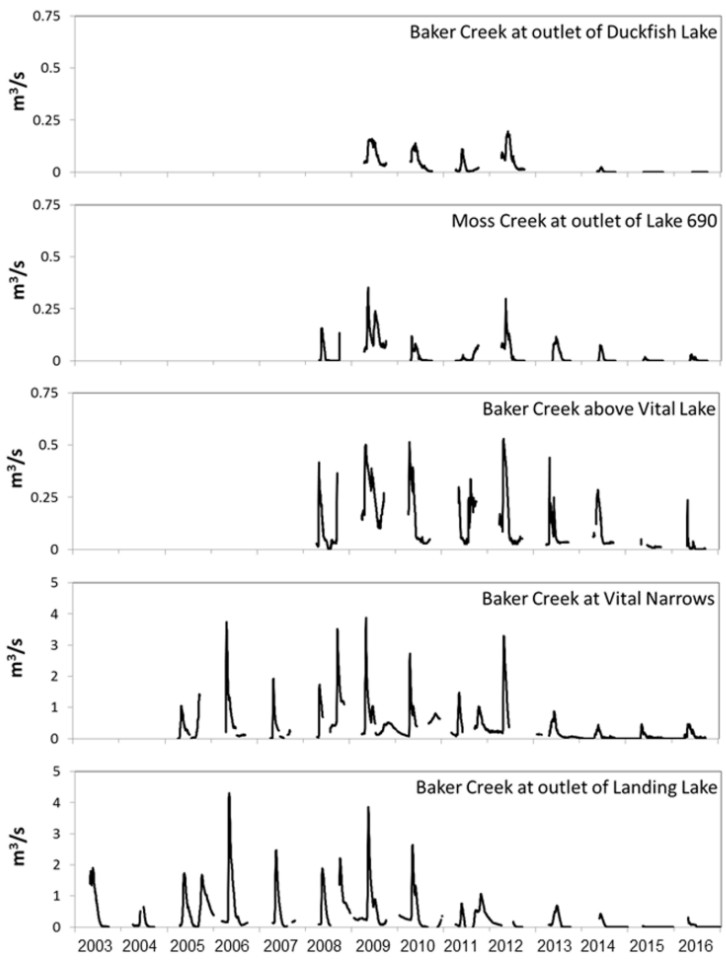


Figure 8: Daily streamflow (2003 – 2016) from the series of nested gauges within the Baker Creek Research Watershed.



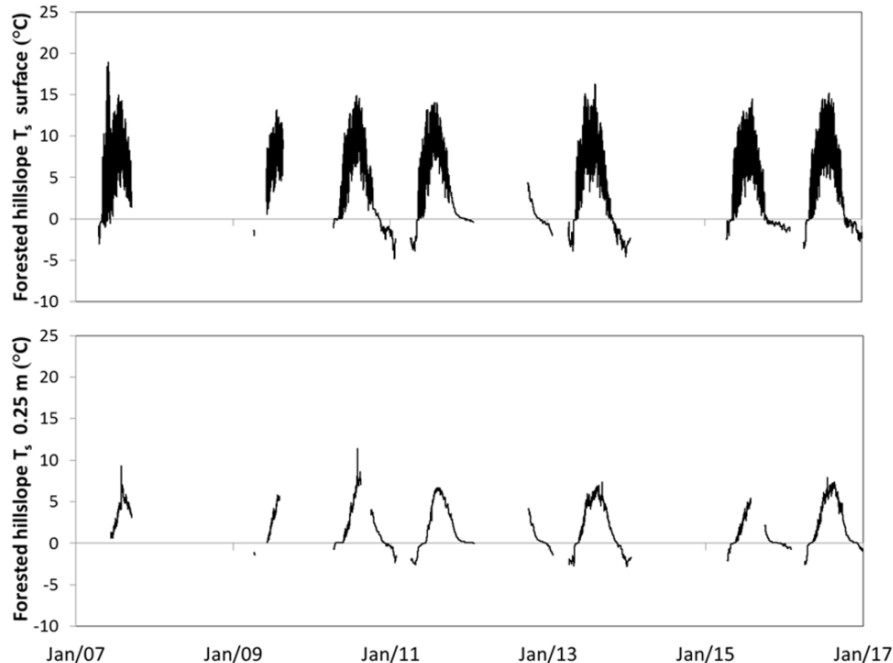

Figure 9:  Half hourly surface and 0.25 m soil temperatures from one of the forested hillslope soil moisture nests
from 2007 to 2016.




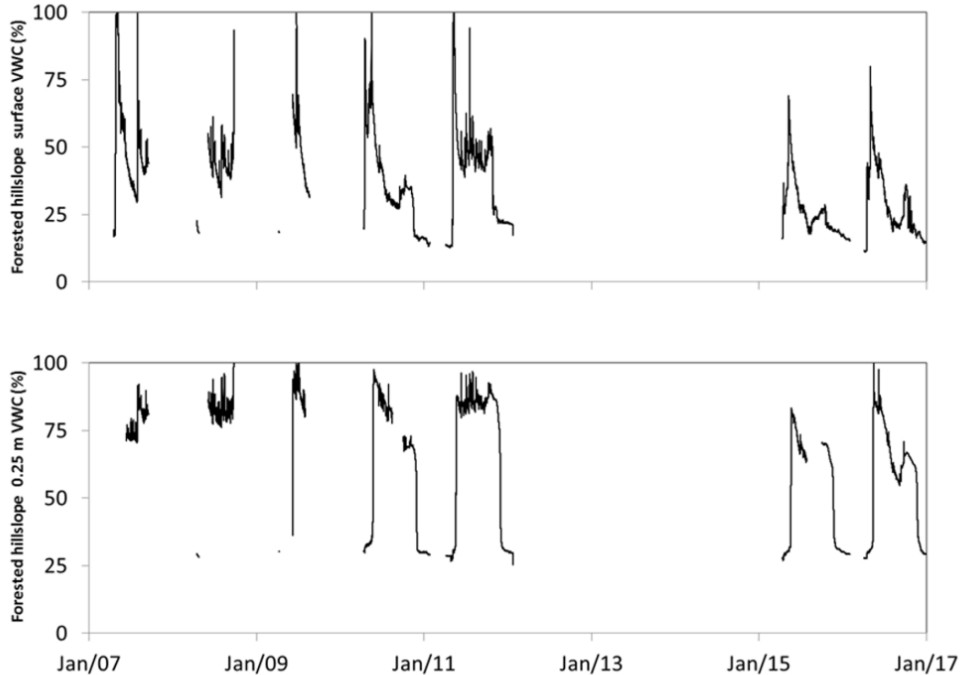

Figure 10: Half hourly soil surface and 0.25 m volumetric water content at the same forested hillslope soil moisture
nest illustrated in Figure 9.



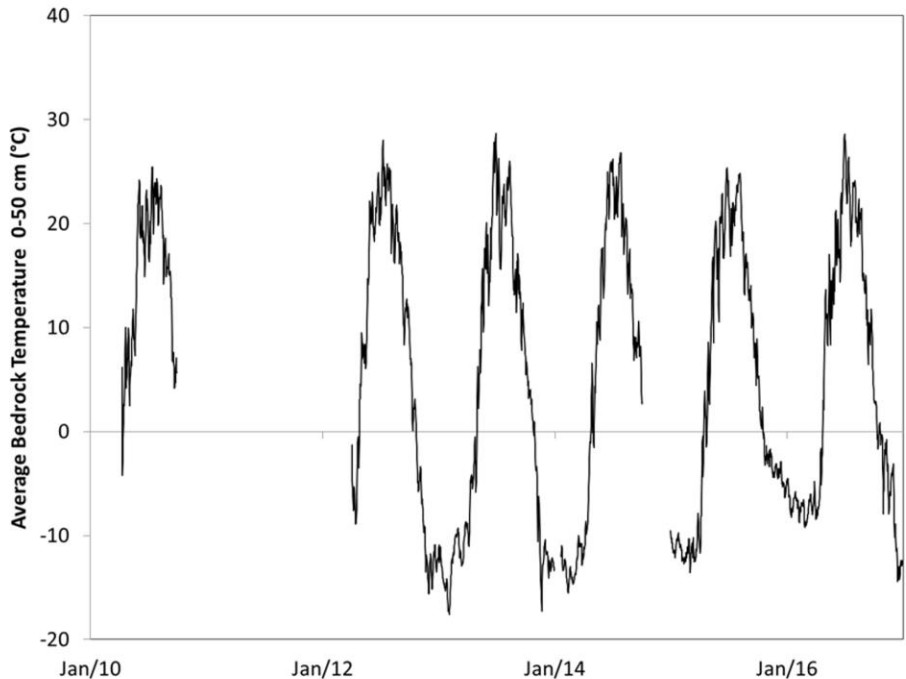

Figure 11: Daily average 0 – 0.5 m bedrock temperatures near the exposed bedrock ridge climate tower from 2010 – 2016.
