# Peer review of "Hydrometeorological Data from Baker Creek Research Watershed, Northwest Territories, Canada"

_Earth System Science Data, 2018_

## Referee Comment (RC1) · Anonymous Referee #1 · 7 Jun 2018

Review of Hydrometeorological Data from Baker Creek Research Watershed, Northwest Territories, Canada. By: Christopher Spence and Newell Hedstrom This paper presents a hydrometeorogical dataset from Baker Creek in the northwest Territories. This dataset is a unique in that provides data for the sparsely sampled northern circumpolar region. Overall this paper is well written and worthy of publication in ESSD subject to minor revisions listed below. A general comment is there needs to be a more systematic description of all sensors and their deployment. The Table 4 list seems to not include everything. A master list would be beneficial. Line 33: clarify why northern flows are important for southern latitudes. Line 36-37: Sentence has two "predict.." Line 38-39: Where are these two stations? Will help to give context to how vast of

an area is being undersampled. Line 40: Do we need multiple stations to glean process information? Line 45: "include that describing" -> describe Line 48-49: ", and so represent. . .." This portion of sentence awkward Line 48-50: this sentence is repetitive to previous discussion Line 55-59: How big is the basin? 155km2 or 165 km2 Line 73: "can be used to characterize" -> characterize Line 75: is there no way of correcting for undercatch as this may be important. If the runoff ratio in Line 79 using the unadjusted precip? Without correcting for undercatch these numbers may be quiet off. At least discuss implications of this Line 87: Two landsat images were used from different years and different points in the phenology. Please elaborate on the challenges on this. How did the classification deal with this difference. Line 120: and fluxes "are" calculated. . ... Line 123: source for oxygen extinction correction? Line 124: here and elsewhere "dataset" rather than "data set" Line 129: what are "sound" conditions? Line 141-143: this sentence is complex. Please simplify 143-144: these albedos values are for the lake surface? Line 147-148: These are annual values even though the dataset is for non-winter periods? Line 149-150: monthly wind speed average is not a meaningful number for most. What about some description of wind speed distributions? Line 151-159: this paragraph may be better situated in the introduction as it helps set the stage and describe importance of the dataset Line 164-170: how was SWE calculated. From mean values or did it consider covariance of depth and density? Line 201-203: awkward sentence

Other comments Missing reference for the Pomeroy and Gray 1995 citation, check others Table 2: there is no description of how these soil properties where measured/observed. Please provide in next Climate tower descriptions are rather limited for the amount of data provided. Pictures? Is the landing lake station on a raft? Or shore? Table 4: model number for surface temperature sensor and licor pyranometers? How was roughness and displacement height calculated or observed? No precipitation storage gauge? Any way of determining phase? Figure 7: Remove basin average value and without doing a statistical analysis don't describe trends in the data Figure 9 and 10: These instruments are not included in Table 4. Are there more instruments

that need to be included.

---

## Referee Comment (RC2) · Anonymous Referee #2 · 8 Jul 2018

General comments This is a fantastic dataset resulting from 13+ years of focused field observations in the Baker Creek catchment. A truly unique set of data that – as the authors state – is valuable for advancing hydrological understanding and has applications for engineering challenges in permafrost regions. The data note is very well written, and – I believe – complete with all necessary information (eg instrumentation, dates, locations, needed for the data user). I have no major comments, and only have minor – largely grammatical – comments. I suggest publication of this data note following addressing of said minor comments.

Major comments N/A

[Figure]

Minor comments âĞć Line 12. Use of word 'seasonal'? âĞć Line 14. "include" not "includes" âĞć Line 17. Delete first "from" âĞć Line 29. And Ontario? âĞć Line 33-34. Could you spell out for us what effects on southerly latitudes via the Arctic Ocean? Circulation? âĞć Line 38. Change "2" to "two". âĞć Line 38-39. Only two for the whole Canadian Shield?! Do you mean climates stations or do you mean research catchments? What time series length is needed to be adequate for long term change detection? âĞć Line 44. Change "2000's" to "2000s" âĞć Line 46. Sentence beginning "These data..." doesn't make much sense- suggest changing "These data are the only..." to "These data constitute the only..." âĞć Figure 1. This needs enlarging; the font size on the axes and legend are too small. Perhaps rotate landscape and enlarge the fonts? This would also let us see the catchment details a little better. âĞć Line 117. Delete the semicolon. Should be a comma. âĞć Line 119. Comma after "surface characteristics" âĞć Line 121. Comma after "(Kaimal and Finnigan, 1994)"? âĞć Line 126 to 128. Too many "because"s in this sentence. Reads rather funny. Suggest re-writing. âĞć Line 129. I'm unclear as to what you mean by "or sound conditions" âĞć Line 132. Change "consistent" to "constant" âĞć Line 153. Remove apostrophe from 1940s. Same for Line 158. âĞć Line 155. Decreased by 27% âĞć Line 159. Additional or alternative reference for the 2014 being the most extensive? Eg.??: Walker, XJ et al (2018) Cross-scale controls on carbon emissions from boreal mega-fires of the Northwest Territories, Canada. Global Change Biology, https://doi.org/10.1111/gcb.14287. âĞć Line 183. Comma after "(07SB013)" âĞć Line 196. This line doesn't read well, maybe there is a word missing somewhere. Suggest re-reading and editing. âĞć Line 206. Here you have written "Water Survey of Canada" in full, but in the previous paragraph you used the acronym. Be consistent: either use the full name or the acronym once defined. âĞć Line 215. Suggest putting in a date here, at which this text is written. Ie. "not experienced outflow since 2014 up until the time of writing (xx 2018..)". Is this still the case? âĞć Line 226. Change semicolon to colon. âĞć Line 272. Delete "and" before "Stefan Goodman"? âĞć Figures. Better consistency with font sizes, types and labelling between the figures. Eg. noticeably different font sizes between Figures 9

and 10, and others. In Figure 5, you write "Jan-08" with a hyphen, while in Figure 9 it is "Jan/07" with a slash. Check inconsistencies among figures, and edit to improve.

---

## Referee Comment (RC3) · Anonymous Referee #3 · 30 Jul 2018

I have read the manuscript by Spence and Hedstrom outlining the data collected for Baker Creek, Northwest Territories, Canada.

I commend the authors for pulling together 13 years of data from such a challenging environment. The data set for this location is absolutely unique and can be the basis for future research for years to come. The inclusion of eddy covariance data is particularly impressive. I have read the others reviewers comments as I note that I am late in my review and wanted to add value-added statements. I did find that some of the instrumentation details (manufacturer, etc) could be strengthened but I worry that time has made some of this information challenging to determine. The data is well sorted in

a repository and easy to decipher.

My only additions are if there is additional soil profile information, this would be useful. In addition, some of the language is a bit casual. I've never been a fan of the word 'gleaned', although it is a word.

---

## Author Comment (AC1) · 31 Aug 2018

**Reviewer #1**

Review of Hydrometeorological Data from Baker Creek Research Watershed, Northwest Territories, Canada. By: Christopher Spence and Newell Hedstrom This paper presents a hydrometeorogical dataset from Baker Creek in the northwest Territories. This dataset is a unique in that provides data for the sparsely sampled northern circumpolar region. Overall this paper is well written and worthy of publication in ESSD subject to minor revisions listed below.

A general comment is there needs to be a more systematic description of all sensors and their deployment. The Table 4 list seems to not include everything. A master list would be beneficial.

**A more thorough description of the sensors and how they were deployed has been added to the text. We have reviewed other papers within ESSD that include tables comparable to our Table 4 and find that the only information ours does not include that others do are the operating ranges, accuracy and sensitivity. Readers, if they are that interested, can easily find this information in operating manuals of these sensors with the data that is provided in the table.**

Line 33: clarify why northern flows are important for southern latitudes.

**This has been done by editing the end of this paragraph.**

Line 36-37: Sentence has two "predict.."

**Fixed.**

Line 38-39: Where are these two stations? Will help to give context to how vast of an area is being undersampled.

**During the process of double checking where these are, it was found that there are actually no stations that meet this requirement. The text has been changed.**

Line 40: Do we need multiple stations to glean process information?

**Not necessarily, but it sure helps in understanding how water is transferred through these landscapes. The text has been adjusted to reflect this.**

Line 45: "include that describing" -> describe

**Fixed. Thanks.**

Line 48-49: ", and so represent: : :." This portion of sentence awkward

**It has been removed.**

Line 48-50: this sentence is repetitive to previous discussion

**This has been retained, because the awkward second part of the previous sentence has been removed.**

Line 55-59: How big is the basin? 155km2 or 165 km2

**We believe the text is clear. The basin at its outlet at Great Slave Lake is 165 km2. The observation focus is on the upper reaches. This area is defined as that above the WSC gauge, which is 155 km2.**

Line 73: "can be used to characterize" -> characterize

Fixed.

Line 75: is there no way of correcting for undercatch as this may be important.

**There is, but published data correcting for undercatch corrections stops in 2014, so we report the unadjusted value. We explicitly state it is unadjusted so readers are aware.**

Is the runoff ratio in Line 79 using the unadjusted precip? Without correcting for undercatch these numbers may be quiet off. At least discuss implications of this

**We have included a statement saying that this runoff ratio may be an overestimate because we used unadjusted precipitation values in the calculation.**

Line 87: Two landsat images were used from different years and different points in the phenology. Please elaborate on the challenges on this. How did the classification deal with this difference.

**A sentence: "Using two images, one prior to and one after leaf-out, assists in differentiating vegetation." has been added. It is actually not a challenge, but helps reduce errors between similar land cover types, particularly those with shrubs.**

Line 120: and fluxes "are" calculated:

**Fixed.**

Line 123: source for oxygen extinction correction?

**Blanken et al., 2011 has been added to the text and reference list.**

Line 124: here and elsewhere "dataset" rather than "data set"

**Thanks. Fixed.**

Line 129: what are "sound" conditions?

**This has been rephrased to: "or coats sensors to prevent accurate readings."**

Line 141-143: this sentence is complex. Please simplify

**The sentence has been split in two.**

Line 143-144: these albedos values are for the lake surface?

**Yes. We have added "of the lake surface".**

Line 147-148: These are annual values even though the dataset is for non-winter periods?

**This value was derived from data from the Meteorological Service of Canada and this is now in the text.**

Line 149-150: monthly wind speed average is not a meaningful number for most. What about some description of wind speed distributions?

**The details about mean wind speed, as well as high winds (>10 m/s) are now provided.**

Line 151-159: this paragraph may be better situated in the introduction as it helps set the stage and describe importance of the dataset

**Some of the broad background content has been moved to the introduction, with specific details remaining in this section.**

Line 164-170: how was SWE calculated. From mean values or did it consider covariance of depth and density?

**SWE was calculated following Pomeroy and Gray (1995) as the product of the density and depth of the snowpack. The data shows that snowpacks are rarely over 60 cm deep, and as suggested by Pomeroy and Gray (1995) shallow snowpacks do not exhibit covariance between depth and density. This information has been added to the text.**

Line 201-203: awkward sentence

Other comments:

Missing reference for the Pomeroy and Gray 1995 citation, check others

**Added, and double checked.**

Table 2: there is no description of how these soil properties where measured/observed. Please provide in next

**Text has been added to describe how these soil properties were calculated, and references provided.**

Climate tower descriptions are rather limited for the amount of data provided. Pictures? Is the landing lake station on a raft? Or shore?

**We have added both a description of each of the towers and pictures in the form of a new Figure 2.**

Table 4: model number for surface temperature sensor and licor pyranometers?

**These have been added to Table 4.**

How was roughness and displacement height calculated or observed?

**Text has been added to the caption of Table 4 to provide a citation to Oke (1987) that provides the equations for these heights.**

No precipitation storage gauge? Any way of determining phase?

**No. Text has been added to state this explicitly. "This type of gauge is not designed to measure solid precipitation. Hence, the dataset includes only rainfall data and not precipitation data. "**

Figure 7: Remove basin average value and without doing a statistical analysis don't describe trends in the data

**We have removed the line representing the average value, and don't describe trends in the data in the text.**

Figure 9 and 10: These instruments are not included in Table 4. Are there more instruments that need to be included?

**As described in the text at these soil moisture nests are not associated with the climate towers. The sensors and their deployment are described in the text at the beginning of Section 7.**

Reviewer #2

General comments This is a fantastic dataset resulting from 13+ years of focused field observations in the Baker Creek catchment. A truly unique set of data that – as the authors state – is valuable for advancing hydrological understanding and has applications for engineering challenges in permafrost regions. The data note is very well written, and – I believe – complete with all necessary information (eg instrumentation, dates, locations, needed for the data user). I have no major comments, and only have minor – largely grammatical – comments. I suggest publication of this data note following addressing of said minor comments.

Minor comments

Line 12. Use of word 'seasonal'?

**This sentence has been rephrased.**

Line 14. "include" not "includes"

**Thanks. Fixed.**

Line 17. Delete first "from"

**Thanks. Fixed.**

Line 29. And Ontario?

**There is no subarctic region in Ontario's Canadian Shield. Only the Hudson Bay Lowlands.**

Line 33-34. Could you spell out for us what effects on southerly latitudes via the Arctic Ocean? Circulation?

**Reviewer #1 also asked this question. Please see our response to reviewer #1 for our response.**

Line 38. Change "2" to "two".

**This sentence has been changed, and the 2 no longer exists.**

Line 38-39. Only two for the whole Canadian Shield?! Do you mean climates stations or do you mean research catchments? What time series length is needed to be adequate for long term change detection?

**Reviewer #1 also asked us to confirm this. The new sentence is "There are no long term climate stations in the region for which there are up to date precipitation data adequate for long term change detection". Adequate for long term change detection means having an adjusted homogenous record.**

Line 44. Change "2000's" to "2000s"

**Done.**

Line 46. Sentence beginning "These data: : :" doesn't make much sense- suggest changing "These data are the only: : :" to "These data constitute the only: : :"

**Thanks. We chose "These data constitute the only …."**

Figure 1. This needs enlarging; the font size on the axes and legend are too small. Perhaps rotate landscape and enlarge the fonts? This would also let us see the catchment details a little better.

**We have decided to split Figure 1 into three separate figures (one each for blueline and instrumentation, land cover, and elevation) to address the reviewers comments.**

Line 117. Delete the semicolon. Should be a comma.

**Typo. Thanks. Fixed.**

Line 119. Comma after "surface characteristics"

**No. We feel one is not required.**

Line 121. Comma after "(Kaimal and Finnigan, 1994)"?

**Thanks. Added.**

Line 126 to 128. Too many "because"s in this sentence. Reads rather funny. Suggest rewriting.

**Done. The sentence has been broken up and rephrased.**

Line 129. I'm unclear as to what you mean by "or sound conditions"

**We addressed this via one of reviewer #1's comments.**

Line 132. Change "consistent" to "constant"

**We respectfully disagree, because it is not constant. It varies.**

Line 153. Remove apostrophe from 1940s. Same for Line 158.

**Done.**

Line 155. Decreased by 27%

**The word has been added.**

Line 159. Additional or alternative reference for the 2014 being the most extensive? Eg.??: Walker, XJ et al (2018) Cross-scale controls on carbon emissions from boreal mega-fires of the Northwest Territories, Canada. Global Change Biology, https://doi.org/10.1111/gcb.14287.

**Excellent suggestion. Thanks.**

Line 183. Comma after "(07SB013)" Line 196. This line doesn't read well, maybe there is a word missing somewhere. Suggest re-reading and editing.

**This has been changed.**

Line 206. Here you have written "Water Survey of Canada" in full, but in the previous paragraph you used the acronym. Be consistent: either use the full name or the acronym once defined.

**We introduce the acronym in Section 2, and now use it throughout once defined.**

Line 215. Suggest putting in a date here, at which this text is written. Ie. "not experienced outflow since 2014 up until the time of writing (xx 2018..)". Is this still the case?

**Yup.  Still dry.  We have added 2018 to the text.**

Line 226. Change semicolon to colon.

**Changed.**

Line 272. Delete "and" before "Stefan Goodman"?

**Deleted.**

Figures. Better consistency with font sizes, types and labelling between the figures. Eg. noticeably different font sizes between Figures 9 and 10, and others. In Figure 5, you write "Jan-08" with a hyphen, while in Figure 9 it is "Jan/07" with a slash. Check inconsistencies among figures, and edit to improve.

**All the figures have been redrafted so that font size and date are the same throughout the paper.**

**Reviewer #3**

I have read the manuscript by Spence and Hedstrom outlining the data collected for Baker Creek, Northwest Territories, Canada. I commend the authors for pulling together 13 years of data from such a challenging environment. The data set for this location is absolutely unique and can be the basis for future research for years to come. The inclusion of eddy covariance data is particularly impressive. I have read the others reviewers comments as I note that I am late in my review and wanted to add value-added statements.

I did find that some of the instrumentation details (manufacturer, etc) could be strengthened but I worry that time has made some of this information challenging to determine.

**We have addressed the concerns of the reviewers by ensuring that all the manufacturer and model details are included.**

The data is well sorted in a repository and easy to decipher.

**Thanks.**

My only additions are if there is additional soil profile information, this would be useful.

**Sorry.  We do not have additional soil profile data.**

In addition, some of the language is a bit casual. I've never been a fan of the word 'gleaned', although it is a word.

**We have removed the word 'gleaned'.  A thorough proof read has also formalized some of the language.**